# Comparative Study of Single-Trait and Multi-Trait Genomic Prediction Models

**DOI:** 10.3390/ani14202961

**Published:** 2024-10-14

**Authors:** Xi Tang, Shijun Xiao, Nengshui Ding, Zhiyan Zhang, Lusheng Huang

**Affiliations:** National Key Laboratory for Swine Genetic Improvement and Germplasm Innovation, Jiangxi Agricultural University, Nanchang 330045, China; tangxi1997@foxmail.com (X.T.); shjx_jxau@hotmail.com (S.X.); dingyd2005@hotmail.com (N.D.)

**Keywords:** genomic selection, multi-trait model, single-trait model, integrated breeding, GEBV

## Abstract

**Simple Summary:**

This study investigates the effectiveness of multi-trait models in genomic selection, which considers the interactions between different traits to improve breeding accuracy. Using simulations based on data from 5000 individuals, we compared multi-trait models with single-trait models across varying population sizes and genetic correlation levels. Our findings show that multi-trait models generally offer better breeding accuracy, especially when the traits have sufficient heritability. However, they also require more modeling time and computational resources. Interestingly, the benefits for traits with low heritability are limited, regardless of genetic correlation. We suggest adopting customized breeding strategies to optimize both efficiency and accuracy.

**Abstract:**

Conventional genomic selection models trait individually, neglecting complex trait interactions. Multi-trait models address this by considering genetic correlations, thus improving breeding value accuracy. Despite their theoretical benefits, quantifying these models’ breeding advantages across genetic backgrounds is essential. This study evaluates the benefits of multi-trait models under varying population sizes and three levels of genetic correlations (low, medium, high) using simulations based on 50 K chip data from 5000 individuals. In equal heritability scenarios, the multi-trait GBLUP model consistently outperforms single-trait models, with breeding advantages increasing with heritability. For example, with a reference population of 4500, improvements range from 0.3% to 4.1%. Notably, trait combinations with low heritability are insensitive to changes in genetic correlation, with gains remaining ≤ 0.1% across different genetic correlations under low heritability conditions. In differing heritability scenarios, the multi-trait model’s benefits vary, particularly enhancing low-heritability traits when paired with high-heritability ones. Additionally, modeling time increases as genetic correlation decreases. The results of this study indicate that multi-trait models improve breeding accuracy but require more modeling time and place higher demands on algorithms and software. We recommend breeding strategies tailored to different phenotypes and genetic backgrounds to balance efficiency and accuracy.

## 1. Introduction

With the continuous development of sequencing technology and the reduction of gene chip costs year by year, the genomic selection (GS) method has become a research hotspot and cutting-edge technology in the field of breeding since it was proposed by Meuwissen et al. [1], bringing revolutionary improvements to livestock and poultry production capacity. Important economic traits of pigs, such as the offspring number, daily weight gain, feed conversion rate, backfat thickness, and meat quality, typically receive greater emphasis in breeding. Research results demonstrate a substantial genetic correlation among several economic qualities, with certain correlations above 0.8 [2,3,4,5]. An extensive examination of the genetic relationships between various traits might offer important guidance for genetic improvement strategies in animal breeding [6].

The single-trait model remains the mainstream approach in GS, where modeling is based on individual traits, and specific weights are assigned to the breeding values of different traits. The selection index is then calculated as the sum of these weighted values, and candidate groups are selected based on the magnitude of the selection index [7]. This method has achieved remarkable results in several cases, but its inherent limitation is that it cannot handle the genetic relationships between multiple traits, resulting in potential genetic correlation information being wasted [8,9]. In addition, single-trait models require multiple independent modeling processes and the review and documentation of numerous result files when dealing with multiple traits. This repetitive procedure increases the operational costs for breeding staff [10]. The multi-trait model can partially alleviate these issues. There have been several multi-trait applications within GS frameworks to address these issues, utilizing various implementations depending on the breeding objectives and traits of interest. Multi-trait genomic selection models go beyond treating each trait independently by incorporating the genetic correlations between traits within a single modeling process. This approach not only captures shared genetic signals more accurately but also allows for a systematic handling of genetic patterns across traits, thereby improving prediction accuracy and statistical power. Moreover, multi-trait models provide better parameter estimates and reduce biases associated with single-trait selection, enabling breeders to identify and select individuals with desirable trait combinations more accurately and effectively [11,12].

While numerous studies have already indicated the considerable potential of multi-trait models in genetic improvement programs [12], there is still a lack of explicit comparisons between single-trait and multi-trait analyses in the literature. Specifically, a systematic evaluation and quantification of the breeding advantages of these models across diverse genetic backgrounds are needed. In practical genomic selection breeding, a training population (core group) and a candidate population are typically involved. A genomic selection model is first established using the training population and then applied to the candidate population to estimate breeding values, which are subsequently used for selection. Accordingly, this study constructed an F1 population to simulate this breeding scenario, with 5000 parent individuals as the training population and their F1 progeny as the candidate population. The accuracy of breeding value estimation for the F1 generation was employed as the primary evaluation criterion. This assessment aims to provide foundational insights for multi-trait model research and their practical application in the industrialization of livestock breeding.

## 2. Materials and Methods

### 2.1. Ethics Statement

All samples used in this experiment were obtained from two pig-breeding companies in Guangdong and Jiangxi provinces, China (Wens Foodstuff Group Co., Ltd. (Yunfu, China) and Aonong Biological Technology Group Co., Ltd. (Nanchang, China)). Jiangxi Agricultural University provided breeding technical services to these companies, and the use of experimental materials was authorized by the respective company authorities. All procedures involving animals follow the Guide for the Care and Use of Laboratory Animals (GB/T 27416-2014, Laboratory Animal Establishment-General Requirements for Quality and Competence) and have been approved by the National Standards of People’s Republic of China and the Ethics Committee of Jiangxi Agricultural University.

### 2.2. Animals

This study selected a total of 5000 randomly sampled individuals from three livestock populations, comprising 1919 Duroc (361 boars and 1558 sows), 2328 Landrace (86 boars and 2242 sows), and 753 Yorkshire (53 boars and 700 sows). Ear samples were collected from all individuals for 50 K chip sequencing. The basic information for the three populations is presented in Table 1 and Appendix A.

### 2.3. Genotyping

To preserve genuine linkage disequilibrium and population characteristics in the simulated genome that followed, this study employed the CC1 PorcineSNP50 BeadChip (51,368 SNPs) to genotype the above 5000 individuals. The marker density and accuracy of the CC1 chip have been previously described in our earlier research [13]. The raw SNP data were then quality-controlled using PLINK v1.9 [14], excluding individuals with a call rate < 95%, SNPs with SNP call rates < 95%, minor allele frequencies < 5%, and SNPs that do not satisfy the Hardy–Weinberg equilibrium (*p* < 10^−5^). Ultimately, 38,101 SNPs and 5000 individuals passed quality control and were retained for further statistical analysis.

### 2.4. Simulate F1 Genotypes and Phenotypes

The genotypes of the F1 group were simulated based on the genotypic data of 5000 real parents. Haplotypes for the parental genotypes were constructed using SHAPEIT v4.2.1 software [15]. Chromosomes were randomly sampled from the male and female gamete pools for recombination to construct the offspring genome. Each chromosome in the newly recombined diploid was simulated with 4–6 random crossover events. Since only one generation was simulated, no consideration was given to the effects of mutations or other disturbances.

To comprehensively evaluate the multi-trait model, this study simulated nine trait combinations with different heritabilities (0.1, 0.3, 0.5) and genetic correlations (0.2, 0.5, 0.8). Each combination included all possible pairwise combinations of the two traits, each controlled by 500 quantitative trait loci (QTLs), with linkage disequilibrium (LD) governing their genetic correlations.

Taking the A1, B1 phenotype combination as an example, both traits have a heritability of 0.1, and the genetic correlation is 0.2. Firstly, 500 SNPs were randomly sampled from the loci separated from the parental population of 5000 as the QTLs for the A1 phenotype. The effects of these 500 QTLs are sampled from a gamma distribution with a shape parameter of 0.4 and a scale parameter of 2/3, and they are randomly assigned positive or negative effects. The simulated QTL effects were multiplied by the allelic genotypes (0, 1, or 2) of the causative loci and summed to obtain the true breeding values (TBVs) for all individuals. Subsequently, the breeding value variance (additive genetic variance) for the population was computed. The residual variance (environmental variance) was calculated using the formula h^2^ = V_g_/(V_g_ + V_e_), and then the residuals are sampled from the standard normal distribution N (0, V_e_). The residuals were added to the breeding values to obtain the phenotypic values for the A1. For the B1 phenotype, SNPs with LD = 0.2 were selected around each QTL from the A1 phenotype to serve as the QTLs for the B1 phenotype. Each pair of QTLs was endowed with identical QTL effects to ensure a genetic correlation of 0.2 between the two traits. The above steps were then repeated to generate the TBV and phenotype for the B1 (Figure 1).

### 2.5. Construction of Single-Trait Models and Multi-Trait Models

Use the following mixed linear model to estimate the variance component of the single-trait model and calculate the breeding value:y=1u+Zg+e
where y is the phenotype vector for individuals with genotypic information in the reference population, u is the population mean, 1 is a unit vector with all elements equal to 1, g is the vector of additive genetic effects (breeding values), following a normal distribution N (0, Gσa2), where σa2 is the additive genetic variance, and G is the marker-based genomic relationship matrix constructed as G = (MM^’^)/(∑2p_i_(1 − p_i_)), where pi is the minor allele frequency at the i-th locus, M is an n * m matrix (n is the sample size, m is the number of markers), Z is the structure matrix corresponding to g, e is the residual, following N (0, Iσe2), where σe2 is the residual variance, and I is an n * n identity matrix.

The expression for the multi-trait model is as follows:$$\left[\begin{array}{c} y_{1}\\ y_{2} \end{array}\right]=\left[\begin{array}{cc} I_{1} & 0\\ 0 & I_{2} \end{array}\right]\left[\begin{array}{c} u_{1}\\ u_{2} \end{array}\right]+\left[\begin{array}{cc} Z_{1} & 0\\ 0 & Z_{2} \end{array}\right]\left[\begin{array}{c} g_{1}\\ g_{2} \end{array}\right]+\left[\begin{array}{c} e_{1}\\ e_{2} \end{array}\right]$$
where $\left[\begin{array}{c} y_{1}\\ y_{2} \end{array}\right]$ represents two phenotypic vectors with genetic correlations, I_1_ and I_2_ are identity matrices, $\left[\begin{array}{c} u_{1}\\ u_{2} \end{array}\right]$ represents the overall means for the two phenotypes, $\left[\begin{array}{c} g_{1}\\ g_{2} \end{array}\right]$ represents the genotypic breeding values for the two phenotypes, following a normal distribution N (0, G⊗M), where $M=\left[\begin{array}{cc} \sigma_{g1}^{2} & \sigma_{g12}\\ \sigma_{g12} & \sigma_{g2}^{2} \end{array}\right]$ is the additive variance–covariance matrix for the two traits, G is the genomic relationship matrix, and Z_1_ and Z_2_ are structure matrices for g_1_ and g_2_. $\left[\begin{array}{c} e_{1}\\ e_{2} \end{array}\right]$ represents the residuals for the two traits, following N (0, I⊗R), where I is an identity matrix, and $R=\left[\begin{array}{cc} \sigma_{e1}^{2} & \sigma_{e12}\\ \sigma_{e12} & \sigma_{e2}^{2} \end{array}\right]$ is the residual variance–covariance matrix for the two traits.

In this study, both single-trait and multi-trait models were constructed using the BLUPF90 software (version 1.0.1) [16]. The estimation of variance–covariance matrices was carried out by invoking the AIREMLF90 module, and breeding values were computed based on the BLUPF90 module. Ultimately, the performance of the models was assessed using the square of the correlation (coefficient of determination) between the predicted breeding values (PBVs) and the true breeding values (TBVs) for 2000 individuals in the F1 generation.

### 2.6. Two Evaluation Scenarios of the Multi-Trait Model

In order to achieve a comprehensive assessment of the multi-trait model, we designed two distinct breeding scenarios.

In scenario 1, we simulated nine different phenotype combinations with varying heritabilities and genetic correlations (Figure 2a). In this scenario, the heritabilities of the two traits in each combination were equal. Additionally, we explored the impact of different reference population sizes on the performance of the multi-trait model (Figure 2b). In the design of different reference population sizes, we strictly controlled the ratio of male to female parents at 1:24, keeping the number of offspring constant at 12. Subsequently, 2000 individuals were randomly selected from the F1 population as the candidate population. The accuracy of PBVs compared to TBVs was then computed for each candidate population size, with each population size simulation repeated 50 times.

In scenario two, we further simulated six phenotype combinations (Figure 3). In this scenario, we introduced two different levels of heritability differences and investigated the advantages of the multi-trait model over the single-trait model at a reference population size of 4500. This scenario was also simulated 50 times for robustness.

## 3. Results

### 3.1. Descriptive Statistical Analysis of Simulation Results

The TBV and phenotypic statistical results for the 5000 reference population (parents) used in the phenotype simulation are presented in Table 2. While keeping TBV constant, traits with different heritabilities were constructed based on three levels of environmental variance. As the heritability increased, all statistical results in the phenotype began to approach the true breeding values, which to some extent showed that the phenotypic simulation was trustworthy. This alignment suggests that our simulation accurately reflects the expected behavior of phenotypic traits under different heritability conditions, providing a foundation for subsequent analyses.

The genetic correlations between phenotypes in this study were entirely regulated by the linkage disequilibrium (LD) between QTLs. The distribution of QTL pairs and the average LD under three genetic correlation backgrounds are illustrated in Figure 4. To achieve a simulated QTL distribution as close as possible to a uniform pattern, QTLs for the first phenotype in each combination were randomly sampled, while QTLs for the second phenotype were selected based on the first phenotype and specific LD criteria.

### 3.2. Genome Prediction Accuracy

Genomic predictions for the nine simulated combinations in scenario 1 were conducted based on different reference population sample sizes (Figure 5). The results indicate that, across all population sizes and combinations of heritabilities and genetic correlations, the multi-trait GBLUP consistently demonstrated higher predictive potential. Specifically, under the same phenotypic genetic correlation background, as the heritability increased, the improvement in predictive accuracy of the multi-trait model compared to the single-trait model also increased. For a reference population size of 4500 (Table 3), the enhancement range for the multi-trait model across the three levels of genetic correlation was 0.3~2.1%, 0.3~3.5%, and 0.4~4.1%, respectively. However, under the same heritability background, different genetic correlation phenotype combinations exhibited varied performance. Specifically, when the reference group size is 4500 (Table 3), as the genetic correlation between phenotypes increases for medium- and high-heritability phenotypes (h^2^ = 0.3 and 0.5), the improvement range of the multi-trait model is 1.0~2.4% and 2.1~4.1%, while for low-heritability phenotypes (h^2^ = 0.1), as the genetic correlation between phenotypes increases, the improvement in prediction performance of the multi-trait model does not change significantly (0.3~0.4%).

In a set of phenotypic combinations with genetic associations, scenarios where the heritabilities are entirely the same are not commonly encountered in practical breeding. Therefore, in scenario two, we constructed phenotypic combinations with differing heritabilities. The results indicate that the multi-trait model’s gains for phenotypes with two different heritabilities within a combination vary (Figure 6). Specifically, under the same phenotypic genetic correlation background, as the difference in heritability between traits increases, the multi-trait model’s gain becomes more pronounced for the trait with lower heritability in the combination. Conversely, the gain is smaller for the trait with higher heritability in the combination. Across the three levels of genetic correlation, the enhancement range of the multi-trait model for the trait with lower heritability in the combination was 1.0~3.0%, 1.8~4.9%, and 3.0~6.2%, respectively. For the trait with higher heritability in the combination, the enhancement range was 0.6~0.8%, 1.2~1.5%, and 2.2~2.7%. Overall, as the genetic correlation within a combination increases, the multi-trait model’s average gain for the combination becomes positively correlated with the total heritability of the combination.

### 3.3. Time Consumption

In real scenarios of multi-trait combined breeding, the substantial time cost associated with independently modeling each phenotype, including the need for separate script submissions and the generation of multiple outcome files requiring individual review and recording, has long been a criticized issue. Therefore, this study conducted a statistical analysis of the runtime for the multi-trait model in scenario 1 (Table 4). In the BLUPF90 software, the entire genomic prediction process involves two parts: the estimation of the variance or covariance matrix and the subsequent calculation of individual breeding values. These are achieved by invoking the AIREMLF90 module and the BLUPF90 module, respectively. The results indicate that differences in heritability and genetic correlation do not significantly impact the runtime during the calculation of individual breeding values. In this phase, the runtime of the multi-trait model is less than twice the runtime of the single-trait model. However, during the construction of the variance and covariance matrix, the runtime of the multi-trait model increases as the genetic correlation decreases. Even when the genetic correlation is 0.8, the runtime of the multi-trait model remains substantially greater than twice that of the single-trait model.

## 4. Discussion

### 4.1. Concerns of Breeders Regarding Multi-Trait Models

Genomic selection technology has been in development in the field of breeding for many years, and multi-trait combined breeding has become increasingly common. While there are many examples of multi-trait genomic evaluations in both research and commercial settings, much of the research still focuses on independently modeling single traits. This seems peculiar since the majority of published studies consistently suggest that, at least theoretically, multi-trait genomic prediction models can enhance the prediction accuracy of the target trait or save breeding costs by leveraging additional information provided by non-target or auxiliary traits [10,11,17].

Here, we present two possible explanations for the concerns expressed by the breeders: On the one hand, multi-trait models are not universally robust, as some research suggests that these models may not significantly enhance, or could even fail to improve, prediction accuracy [10,18]. The effectiveness of multi-trait models relies on the heritability of individual traits and the genetic correlations between them [19]. However, due to factors such as selection and genetic drift, the expression of the same trait can vary significantly across different populations [20]. To avoid applying multi-trait models to traits without meaningful genetic correlations, it is recommended to first estimate genetic correlations using either genomic data or pedigree information. Once the genetic correlations are determined, multi-trait models should be employed only when strong correlations exist between traits, indicating a potential benefit [11]. Nevertheless, this additional analysis represents an extra task for breeding personnel, who may hesitate to adopt this approach without evidence of substantial genetic improvement.

On the other hand, concerns regarding the feasibility of implementing multi-trait models arise due to differences in computational demands, which can be a crucial factor determining the practical application of model selection. Research indicates that the construction time of multi-trait models is longer than that of single-trait models, requiring a higher number of iterations [10]. This aligns with the results of the present study, where the computational cost of genetic parameters is higher for multi-trait models due to a more detailed decomposition of phenotypic variance during the construction process (Table 4). Additionally, this study encountered cases of non-convergence during modeling for trait combinations with low heritability and insufficient correlation. Apart from phenotypic factors, this undoubtedly places higher demands on commercial software and associated algorithms [21,22]. Therefore, a more comprehensive evaluation of multi-trait models is imperative for guiding breeding practices.

### 4.2. The Role of Heritability and Genetic Correlation in Multi-Trait Models

To quantify the impact of heritability and genetic correlation on the effectiveness of multi-trait models, this study simulated nine phenotypic combinations with varying heritability and genetic correlation backgrounds in scenario 1. The results indicate that for phenotypic combinations with medium- or high-heritability, genetic advantages increase with the rise in genetic correlation among phenotypes. In contrast, for combinations with low heritability, changes in genetic correlation seem to have little effect on genetic gains, aligning closely with the results of numerous other studies [23]. This appears reasonable, considering the fundamental equation of multi-trait models (see Materials and methods), where the essence lies in leveraging additional information from correlated traits while providing a more detailed decomposition of variance components, enhancing the connectivity of additive covariance and residual covariance between traits. However, for combinations with low heritability, the genetic information that a single phenotype can offer is inherently limited. Coupled with the randomness of residual effects, the model’s capacity to absorb additional information becomes exceedingly weak, potentially even absorbing harmful information that adversely affects predictive outcomes. In response to this result, we used an interesting analogy to the role of heritability and genetic correlation in multi-trait models (Figure 7). Although this study does not provide direct evidence that the use of multi-trait models under low heritability conditions weakens predictive accuracy, based on efficiency considerations, we strongly recommend breeders devise distinct breeding strategies for different phenotypes.

### 4.3. Low Heritability Traits Benefit from Correlated High Heritability Traits

Phenotypic combinations simulated in scenario 1 are not commonly encountered in real breeding practices. In fact, it is difficult to find two phenotypes with exactly the same heritability. Therefore, in scenario 2, we introduced two different levels of heritability variation among phenotypic combinations. The results indicate that as the disparity in heritability among phenotypes increases, the multi-trait model exhibits greater gains for traits with lower heritability in the combination, while no such improvement is observed for traits with higher heritability. This characteristic is particularly relevant in livestock and poultry breeding, where many traits of interest exhibit lower heritability, such as specific reproductive traits in pigs, certain disease resistances, and certain carcass traits [5,24,25,26]. These traits show limited genetic gains in single-trait models, leading to slow genetic progress. According to the findings of this study, when jointly modeling traits with high heritability, low-heritability traits can achieve an additional gain of around 5%, which is significant for traits with lower heritability. While these gains may vary for different traits, numerous studies consistently suggest that incorporating information from high-heritability traits can effectively reduce selection bias and standard errors for low-heritability traits in univariate analyses [27,28].

### 4.4. Strengths and Considerations in Simulation Based on Real Sequencing Data

This study’s strength lies in the fact that the entire simulation process is based on real sequencing data, preserving the complexity of the genome and the genuine interactions between genotypes and the environment. Therefore, the simulation of QTL extraction based on real sequencing data holds more biological significance. Additionally, this research utilized three purebred commercial populations as parental groups, aiming to minimize the impact of breed specificity and population selection pressures on the results.

However, using an F1 population rather than a purebred population introduces certain challenges. The F1 population results from crossing distinct parental lines, leading to unique allelic frequencies and linkage disequilibrium (LD) structures that differ from the parent populations. This deviation can impact the accuracy of genomic predictions, as the genetic relationships between the reference and candidate populations in the F1 may not fully represent those within purebred populations. Specifically, the allelic frequency differences and potential deviation from Hardy–Weinberg equilibrium in the F1 population can cause biased estimates of breeding values, thereby affecting the reliability of prediction models. Moreover, changes in LD patterns in the F1 can influence the effectiveness of marker-trait associations, which are crucial for genomic selection.

On the other hand, in the process of constructing the progeny’s diploid genome, this study set the number of recombination occurring on each chromosome to be between four and six. In reality, the number of recombination across the entire genome in one generation is typically around 20 (averaging 1–2 times per chromosome) [29]. The recombination frequency in this study undoubtedly increased the intergenerational interval. This represents a carefully considered trade-off because, in real breeding scenarios, there is an accumulation of data in the reference population, leading to an increase in the average generation interval between the reference and candidate populations [30]. This might explain why the overall accuracy of this study is slightly lower than that of other studies that focus on “one generation”.

In this study, all phenotype combinations include only two traits. On the one hand, based on breeding principles, when selecting n traits simultaneously, the selection response for each trait is theoretically 1/√n compared to selecting a single trait [31]. Therefore, theoretically, selecting two traits simultaneously is considered more advantageous than selecting more than two traits simultaneously, as it can yield greater genetic progress. On the other hand, as the number of traits included in the multi-trait model increases, the variance components it dissects also increase. The theoretical time and computational costs tend to grow exponentially [10], and there is a higher likelihood of encountering convergence issues. However, the practical implications warrant further discussion, making it a potential focus for future research in our group.

### 4.5. The Practicality of the Multi-Trait Model Requires Further Investigation

An unavoidable aspect of the discussion is that in this study, all simulated heritabilities are narrow-sense (considering only additive effects). Although some studies suggest that the predictive abilities of additive and additive-dominance models show little difference under conditions of moderate to high heritability [32], the real scenario is much more intricate. In models that neglect dominance effects, especially under conditions of high heritability, dominance variance often tends to be more absorbed into additive variance [32]. The confounding of variance components may lead to erroneous conclusions and potentially harmful decisions in breeding programs, particularly in populations with a certain level of selection pressure. This may indeed be one of the reasons for the divergent results observed in various studies. On the other hand, the main genetic basis for trait correlation is controversial and may be due to the synergy of gene pleiotropy and linkage disequilibrium [33], and the pleiotropy of genes is difficult to accurately control in simulation data.

It is noteworthy that studies have shown that genetic covariance between traits is not uniformly distributed across the genome. This indicates the presence of different “hotspots” within the genome that significantly influence the genetic correlations between traits. Novel models that exploit this heterogeneity to enhance the prediction accuracy of traits with low genetic correlations have been reported [23]. Further research has shown that considering the genetic architecture of the genome, such as the location and effect size of genes influencing traits and the frequency of these genes across different populations, not only improves multi-trait predictions but also enhances multi-population predictions [34].

In summary, the current research on multi-trait models is still insufficient, and we look forward to future studies addressing this gap, particularly in larger population sizes, with more diverse varieties or lines, increased SNP quantities, and real phenotypes. This could accelerate the application of multi-trait models in the breeding industry.

## 5. Conclusions

This study quantifies the breeding gains of multi-trait models under different genetic parameters based on simulations of offspring and phenotypes using 5000 real 50 K chip data. Specifically, in scenario 1 where phenotype combinations have the same heritability, the multi-trait GBLUP consistently outperforms in all outcomes. As genetic heritability increases, breeding gains also increase, especially when the reference population size is 4500, with average gain fluctuating between 0.3% and 4.1%. Phenotype combinations with low heritability are insensitive to changes in genetic correlation, showing gains of no more than 0.1% across different genetic correlations. In scenario 2, where there is a difference in the genetic heritability between traits, the multi-trait model exhibits varied gains based on the heritability of each trait. The results indicate that when the heritability of two traits is unequal, the multi-trait model has a more significant impact on traits with lower heritability. Additionally, the study addresses the time cost of modeling, revealing that as the genetic correlation between phenotype combinations decreases, the time cost of the multi-trait model in the variance–covariance matrix estimation process gradually increases. Overall, this research demonstrates that, under favorable genetic correlations and heritability backgrounds, multi-trait models have considerable breeding potential. It provides valuable insights into the theoretical foundation and industrial application of multi-trait models in livestock breeding.

## Figures and Tables

**Figure 1 animals-14-02961-f001:**
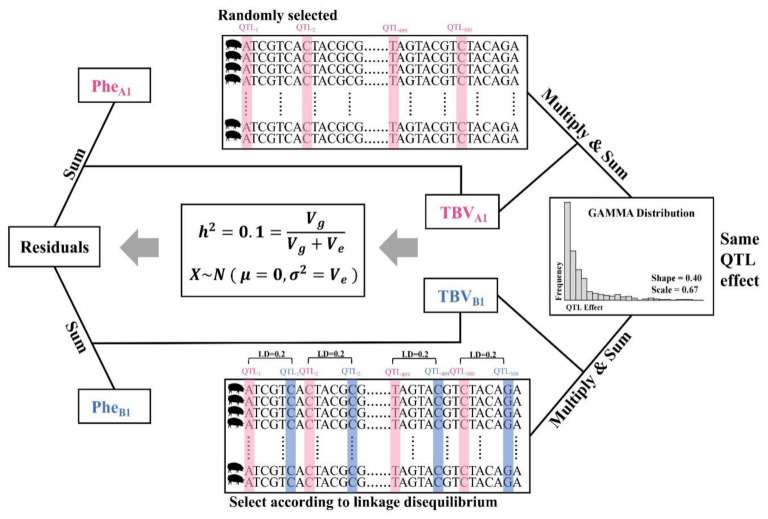
Phenotype Simulation Workflow Chart (Taking A1 and B1 Phenotypes as Examples).

**Figure 2 animals-14-02961-f002:**
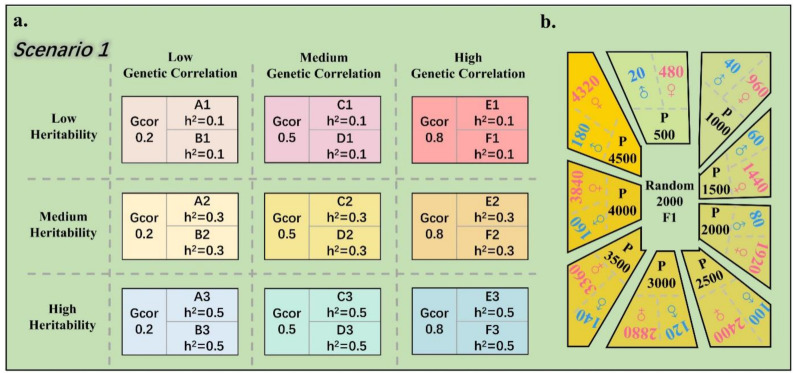
Schematic diagram of scenario 1. In this scenario, we simulated nine different combinations of phenotypes with varying heritabilities and genetic correlations (**a**), and explored the advantages of multi-trait models over single-trait models under different reference population sizes (**b**).

**Figure 3 animals-14-02961-f003:**
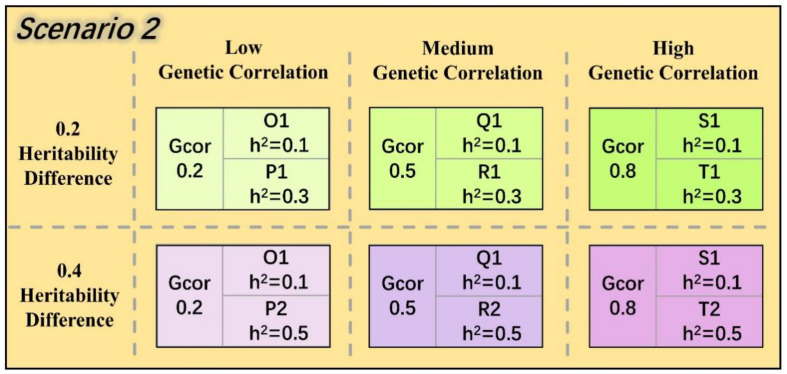
Schematic diagram of scenario 2. In this scenario, we simulated six different phenotype combinations with varying differences in heritability and genetic correlations, and explored the advantages of multi-trait models over single-trait models at a reference population size of 4500.

**Figure 4 animals-14-02961-f004:**
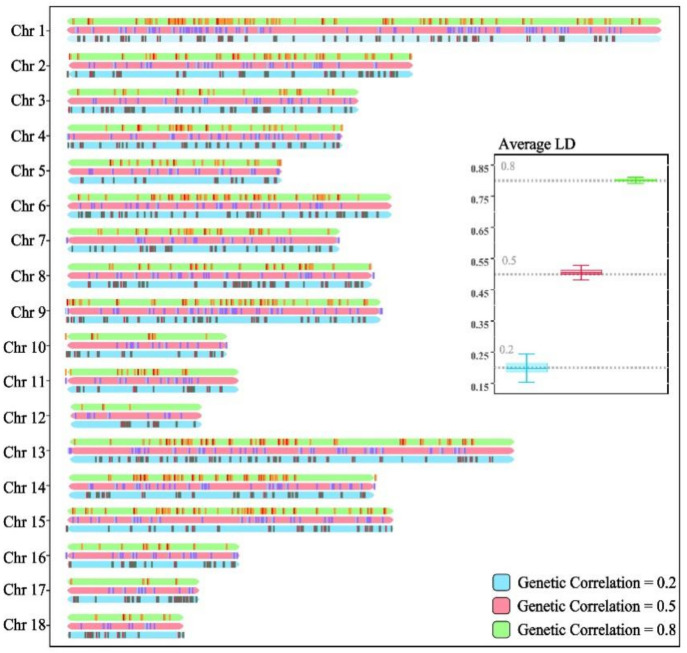
Distribution of QTL pairs of chromosomes in three genetic correlation backgrounds.

**Figure 5 animals-14-02961-f005:**
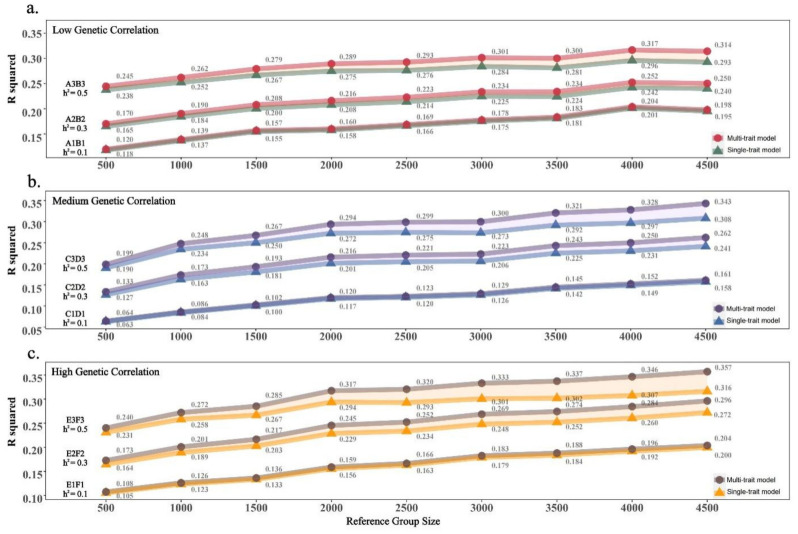
The summarized results for Scenario 1. Each value represents the mean accuracy of predicted breeding values (PBVs) for the two phenotypes (50 simulations results).

**Figure 6 animals-14-02961-f006:**
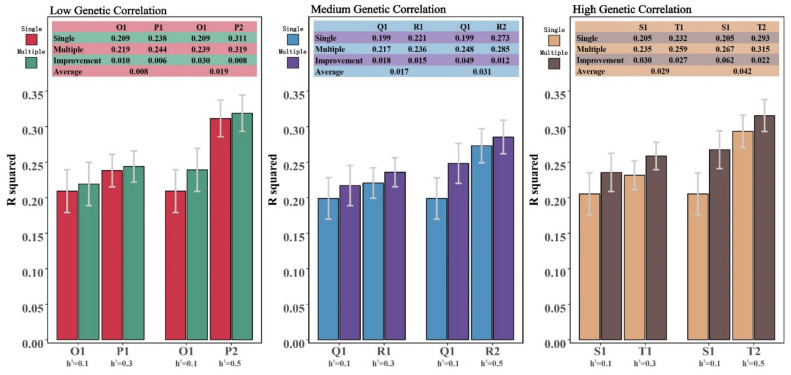
The summarized results for scenario 2 (Reference group size = 4500).

**Figure 7 animals-14-02961-f007:**
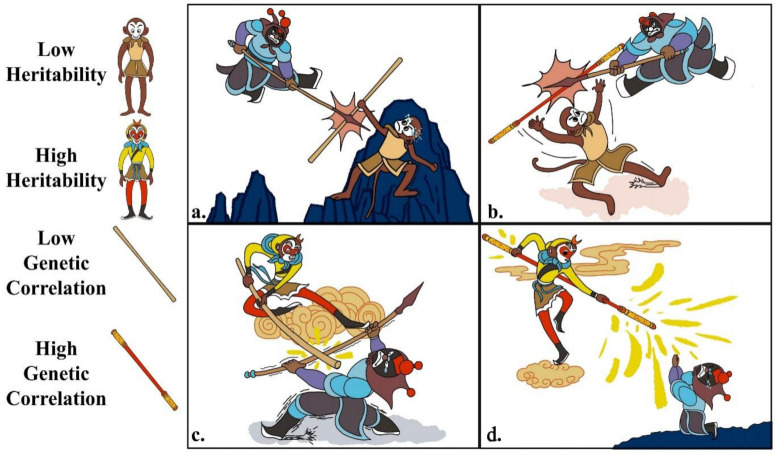
The role of heritability and genetic correlation in multi-trait models. We vividly illustrate the roles of heritability and genetic correlation in multi-trait models using characters from the popular Chinese fantasy novel “Journey to the West.” Heritability, as a crucial attribute of the traits themselves, is represented here by an ordinary monkey for low-heritability traits and the mighty “Monkey King” (Sun Wukong) for high-heritability traits. Genetic correlation, playing a crucial supporting role in multi-trait models, is symbolized by an ordinary staff for low genetic correlation and the powerful “Ruyi Jingu Bang” (Golden Cudgel) for high genetic correlation. When individual abilities are insufficient (low heritability), the assistance provided, whether it is from traits with high or low genetic correlation, does not significantly enhance the overall performance (breeding value prediction accuracy) (**a**,**b**). However, when individual abilities are strong (high heritability), the assistance, whether from traits with high or low genetic correlation, contributes to a certain improvement in overall performance (breeding value prediction accuracy) (**c**,**d**). This improvement is especially notable when combining high heritability and high genetic correlation, resulting in a substantial increase in breeding accuracy (**d**).

**Table 1 animals-14-02961-t001:** Summary information for the three groups.

Group Batch	Breeds	Origins Birthplaces	Breeding Company	Number	Boars	Sows
Group 1	Duroc	Guangdong Province, China	Wens Foodstuff Group Co., Ltd.	1919	361	1558
Group 2	Landrace			2328	86	2242
Group 3	Yorkshire	Jiangxi Province, China	Aonong Biological Technology Group Co., Ltd.	753	53	700
Sum				5000	500	4500

**Table 2 animals-14-02961-t002:** Summary of statistical results of simulated phenotypes of the reference group. SD: standard deviation. CV: coefficient of variation.

Reference Group (5000)		Mean	Median	Variance	SD	CV	Kurtosis	Skewness
TBV	h^2^ = 0.1	2.924	2.829	45.919	6.776	0.261	0.027	0.121
	h^2^ = 0.3							
	h^2^ = 0.5							
Phenotype	h^2^ = 0.1	3.112	3.195	461.185	21.475	0.794	−0.024	0.002
	h^2^ = 0.3	3.020	2.950	153.826	12.403	0.471	0.001	0.013
	h^2^ = 0.5	2.987	2.897	92.278	9.606	0.368	0.019	0.029

**Table 3 animals-14-02961-t003:** The accuracy of predicted breeding values (PBVs) for multi-trait and single-trait models at a reference population size of 4500.

Reference Group Size = 4500	Genetic Correlation = 0.2			Genetic Correlation = 0.5			Genetic Correlation = 0.8		
	A1B1	A2B2	A3B3	C1D1	C2D2	C3D3	E1F1	E2F2	E3F3
Single-trait model (mean/sd)	0.195 (0.028)	0.240 (0.040)	0.293 (0.049)	0.158 (0.059)	0.241 (0.048)	0.308 (0.047)	0.200 (0.029)	0.272 (0.028)	0.316 (0.028)
Multi-trait model (mean/sd)	0.198 (0.029)	0.250 (0.040)	0.314 (0.049)	0.161 (0.060)	0.262 (0.049)	0.343 (0.049)	0.204 (0.029)	0.296 (0.028)	0.357 (0.033)
Improvement	0.003	0.010	0.021	0.003	0.021	0.035	0.004	0.024	0.041

**Table 4 animals-14-02961-t004:** The running speed of single-trait and multi-trait models in the BLUPF90 software during a single simulation, with a reference population size of 4500 and the candidate population is 2000 (average of 50 simulations). *: This result is derived from the time function in a Linux system with a single CPU, single node, and 16 GB of running memory. It represents the actual time elapsed from the beginning to the end of program execution; AIREMLF90: calculate the variance or covariance matrix; BLUPF90: calculate breeding value.

Time(s) *	Genetic Correlation	h^2^ = 0.1		h^2^ = 0.3		h^2^ = 0.5	
		AIREMLF90	BLUPF90	AIREMLF90	BLUPF90	AIREMLF90	BLUPF90
Single-trait model	\	464.29	90.50	454.10	91.66	472.55	93.12
Multi-trait model	0.2	1903.51	141.16	1800.90	135.48	1856.54	133.11
	0.5	1660.48	136.37	1569.59	141.61	1680.50	141.62
	0.8	1557.17	136.69	1500.12	136.39	1571.21	136.75

## Data Availability

The data that support the findings of this study are available from Jiangxi Agricultural University but restrictions apply to the availability of these data, which were used under license for the current study, and so are not publicly available. Data are however available from the authors upon reasonable request and with permission of Jiangxi Agricultural University.

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
