# Peer review of "Comparative Study of Single-Trait and Multi-Trait Genomic Prediction Models"

_animals, 2024, doi:10.3390/ani14202961_

Round 1

Reviewer 1 Report

Comments and Suggestions for Authors

I am writing a review of the manuscript entitled "Comparative study of single-trait and multi-trait genomic prediction models" by Tang et al., submitted to MDPI Animals

This study investigates the advanteages of multi-trait models in different scenarios using simulated data created based on real genotypes. While there is very little novelty in this research, it is important that all of those comparisons are included in a single peer reviewed document. This manuscript will be useful for students learning about multiple-trait models, but will also be benefitial in many discussions of model comparisons. 

It is not clear why the authors created an F1 population, as it adds another layer of complexity to the study. Please, clarify it and include that in the title and introduction/objectives. 

Specific comments

L 50: This sentence is innacurate. There are many multi-trait applications of conventional GS. 

L 50 to 66: This whole paragraph is innacurate. There are many ways to implement MT models in GS

L 67: This justification could be changed to address the lack of published literature with explicit comparisons between single and multi-trait analysis. 

L 109: instead of saying various, please, state the number of comparisons. Are all combinations of heritability and genetic correlation used? (add a link to figure 2a and 3 here to help on that matter). Maybe this part could be moved to L 167 for simplicity.

L 279: This sentence is innacurate. There are many examples of multi-trait genomic evaluations.

L 280: Another innacuracy: most commercial evaluations use multi-trait models when needed

L 292: This advice is not clear. How can genetic parameters of correlations be estimated with single trait analysis? Why not run a pedigree variance component estimation and calculate the genetic correlation?

L 299 - 304: It would be nice to provide a big o notation here for the costs

Figure 7 would be interesting in a presentation or classroom, but I am not sure if it belongs in a scientific paper. I really liked it, and I would like to see it in the final version. I am not sure how the editors will feel about it, but I would recomend to add it as a text box or a visual abstract of the paper. The way it is right now doesn't look very helpful. 

L 404 to 417: Please, add some discussion about the study being in an F1 populatiuon instead of a purebred population. What problems does it bring? How the allelic frequencies and equilibrium would affect the predictions?

Reviewer 2 Report

Comments and Suggestions for Authors

The article presents an assessment of a multi-trait model. This is a rather interesting problem, since despite the growing interest in multi-trait models, there are many controversial issues about their effectiveness, or some methodological points, when and for which traits it is advisable to use them.

The article presents the methods in great detail and well. The authors tried to evaluate multi-trait models and their effectiveness relative to single- trait models in various combinations, taking into account heritability and genetic correlation.

The results are presented in detail and clearly, it is not difficult to understand the data in the figure and tables. The conclusions are well-founded and correspond to the presented results.

Regarding the wishes/comments, I would like to draw the authors' attention to the purpose of the work

Line 69-70: ‘Therefore, the purpose of this study is to comprehensively assess the breeding advantages of multi-trait models…’

Perhaps it is better to use not advantages", but something more neutral, for example, efficiency, capabilities, etc.
